# FUTURE-GB: functional and ultrasound-guided resection of glioblastoma – a two-stage randomised control trial

Puneet Plaha,[1,2] Sophie Camp,[3] Jonathan Cook,[4] Peter McCulloch,[2] Natalie Voets,[5] Ruichong Ma [ID],[1,2] Martin J B Taphoorn,[6,7] Linda Dirven,[6] Matthew Grech-Sollars,[8,9] Colin Watts [ID],[10,11] Helen Bulbeck,[12] Michael D Jenkinson,[13,14] Matthew Williams,[15] Adrian Lim,[16,17] Luke Dixon,[18] Stephen John Price [ID],[19] Keyoumars Ashkan,[20] Vasileios Apostolopoulos,[1] Vicki S Barber,[4] Amy Taylor,[2] FUTURE-GB collaborators, Dipankar Nandi[3,16]

For numbered affiliations see end of article.

**Correspondence to**
Puneet Plaha;
puneet.plaha@ouh.nhs.uk

## ABSTRACT

**Introduction** Surgery remains the mainstay for treatment of primary glioblastoma, followed by radiotherapy and chemotherapy. Current standard of care during surgery involves the intraoperative use of image-guidance and 5-aminolevulinic acid (5-ALA). There are multiple other surgical adjuncts available to the neuro-oncology surgeon. However, access to, and usage of these varies widely in UK practice, with limited evidence of their use. The aim of this trial is to investigate whether the addition of diffusion tensor imaging (DTI) and intraoperative ultrasound (iUS) to the standard of care surgery (intraoperative neuronavigation and 5-ALA) impacts on deterioration free survival (DFS).

**Methods and analysis** This is a two-stage, randomised control trial (RCT) consisting of an initial non-randomised cohort study based on the principles of the IDEAL (Idea, Development, Exploration, Assessment and Long-term follow-up) stage-IIb format, followed by a statistically powered randomised trial comparing the addition of DTI and iUS to the standard of care surgery. A total of 357 patients will be recruited for the RCT. The primary outcome is DFS, defined as the time to either 10-point deterioration in health-related quality of life scores from baseline, without subsequent reversal, progressive disease or death.

**Ethics and dissemination** The trial was registered in the Integrated Research Application System (Ref: 264482) and approved by a UK research and ethics committee (Ref: 20/LO/0840). Results will be published in a peer-reviewed journal. Further dissemination to participants, patient groups and the wider medical community will use a range of approaches to maximise impact.

**Trial registration number** ISRCTN38834571.

## STRENGTHS AND LIMITATIONS OF THIS STUDY

⇒ This is a randomised control trial comparing the quality of life of patients with glioblastoma undergoing standard of care surgery (intraoperative neuronavigation and 5-aminolevulinic acid) versus surgery with the addition of diffusion tensor imaging (DTI) and intraoperative ultrasound (iUS).

⇒ To ensure standardisation and quality control of delivery of the DTI and iUS in the randomised trial, sites will be required to enter a minimum number of patients into an initial IDEAL (Idea, Development, Exploration, Assessment and Long-term follow-up) stage-IIB study prior to commencing recruitment to the randomised trial.

⇒ Patient and public involvement determined the primary outcome measure of deterioration free survival (DFS) (comprising a decline in health-related quality of life, disease progression or death), rather than overall survival. DFS is considered by patients to be most pertinent.

⇒ This trial recruits patients aged 18–70 years who can undergo surgery to maximally resect their glioblastoma.

⇒ There is variability of the iUS machines used by trial sites (sites use machines they are familiar with). However, this reflects real world iUS usage.

## INTRODUCTION

Glioblastoma (GB) is the most frequent and aggressive form of primary brain cancer, with an incidence of 4.64/100 000 persons/year in the UK.[1] Prognosis remains extremely poor with median survival of approximately 15 months,[2] and as the tumour grows, patients experience a progressive decline in health-related quality of life (HRQoL), and caregivers report high levels of distress and carer burden.[3] Resistance to treatment leads to poor survival, with high costs to the patient, relatives, society and the economy.[4 5] Although primary brain tumours represent only 3% of all cancers, a brain tumour reduces life expectancy by an average of 20 years, the highest of any cancer, and accounts for more average years of life lost than any other

cancer.[4 5] GB affects adults in their economic prime, and is a leading cause of death in those under 40 years of age, costing the economy £578 million per year.[4 5] To date, there has been little progress in improving outcomes including quality of life, with many trials failing to show an effect.[6]

Surgery is the mainstay of treatment for GB, but optimum surgical technologies remain unclear. Surgery to resect GB is integral to maximum first-line treatment, with a greater impact on survival than non-operative treatments (radiotherapy and chemotherapy).[7] It improves symptom control, reduces dependence on dexamethasone and increases progression free survival (PFS) and overall survival (OS).[8 9] However, maximising the extent of surgical resection must be balanced against the potential risk of causing neurological deficit, and hence impacting negatively on a patient's ability to tolerate adjuvant treatments.

The desire to achieve a safe, maximal resection, particularly in eloquent regions, has led to an increase in the use of intraoperative imaging. This attempts to eliminate the error produced by brain shift, an inherent problem in navigation systems based on preoperative imaging,[6] to demonstrate residual tumour at operation, and to visualise accurately relevant white matter tracts and tumour margins. Two technologies that facilitate surgical resection intraoperatively are intraoperative ultrasound (iUS) and diffusion tensor imaging (DTI).

1. iUS accommodates for brain shift if it is linked to neuronavigation systems, allowing the surgeon to track tumour resection in real time. iUS permits multiple, real time image acquisitions, and, potentially, if navigated, at each stage, comparison with the preoperative MRI navigation sequence, to evaluate brain shift and residual disease. iUS minimally augments operative time,[6] allowing precise visualisation of tumour resection. It is user friendly, widely available and a pragmatic and cost-effective alternative to intraoperative MRI, which is prohibitively expensive for many UK units. iUS, and more recently navigated iUS, has a long history in brain tumour surgery,[10 11] facilitating/extending resection,[12–16] and improving survival.[17] It has also been evaluated with respect to histology.[18 19] However, there is a learning curve, and image interpretation, especially during resection, can be challenging.[10] iUS demonstrates residual tumour in real time. Indeed, it has been reported that navigated iUS and 5-aminolevulinic acid (5-ALA) provide different information of tumour extent, and when combined, enhance extent of resection.[20] Despite this, there are no randomised trials assessing its efficacy.

2. DTI is a special MRI technique that can identify the location of white matter nerve tracts important for speech/language/visual/motor functions. The location of white matter fibre pathways is the most frequent reason why surgery is halted early, to avoid compromising patient function.[21] DTI is the only method available to visualise functionally important white matter tracts in the vicinity of a tumour before surgery and can be fused with standard intraoperative navigation systems to enable visualisation of the spatial location of the tracts during surgery, allowing removal of tumour in close proximity. The usefulness of DTI in brain tumour surgery has recently been reviewed.[21] Intraoperative visualisation of DTI is reported to contribute to maximising safe resection,[22–24] reducing visual field deficits[25] and predicting long term language problems after surgery.[26] A single centre randomised control trial (RCT), comparing DTI versus no-DTI, showed that DTI led to significantly better gross total resection (GTR) rates, a lower risk of movement loss and improved life expectancy.[27] Furthermore, DTI-informed awake surgery reduced the occurrence and severity of behavioural problems postoperatively, leading to faster recovery, and shorter hospital stay.[28] DTI requires the collection of additional MRI data, specialist software for analysis and detailed knowledge of white matter anatomy and function. In addition, tract visualisation may be restricted where there is peritumoural oedema. As a result, there is only limited data available on the sensitivity and specificity of DTI in GB surgery, particularly with reference to its value as an intraoperative tool and in predicting DFS.

There are wide variations of surgical standard of care across the UK. A survey of all 24 adult UK neurosurgical centres (telephone and email survey conducted in 2018 by the Oxford researchers), showed wide variation in the use of technologies employed during GB resection. While all centres employ standard neuronavigation and 5-ALA, only 75% have access to iUS, 62% to DTI and 16% to an intraoperative MRI scanner. It remains unclear which technologies should be employed intraoperatively, without worsening neurological function. Indeed, most of these technologies are not regularly used for tumour resection, with surgeons unclear of the efficacy of each, and what is the optimum combination. A recent Cochrane review emphasised the lack of high-quality evidence to support the use of any specific intraoperative imaging technology.[29] The National Institute for Health and Care Excellence guidance[3] has suggested that the available range of intraoperative technologies are considered, as appropriate, in addition to standard techniques, for tumour resection.

The Functional and UlTrasound gUided Resection of GlioBlastoma—FUTURE-GB trial was developed in collaboration with the Society of British Neurosurgeons and multiple patient with GB advocate groups to try and address some of the deficits in knowledge regarding the use of additional surgical adjuncts. FUTURE-GB aims to evaluate the impact of DTI and iUS in addition to standard of care techniques with a view to providing high-quality evidence to shape standard practice in the future.

## METHODS AND ANALYSIS
### Trial design and setting
FUTURE-GB is a two-stage trial (figure 1). Patient and public involvement (PPI) actively informed the rationale,

**Figure 1** Trial flowchart. DTI, diffusion tensor imaging; intraop, intraoperative; iUS, intraoperative ultrasound; HRQoL, health-related quality of life; 5-ALA, 5-aminolevulinic acid; IDEAL, Idea, Development, Exploration, Assessment and Long-term follow-up; MOCA, Montreal Cognitive Assessment; MRC, Medical Research Council

design and development of the protocol and patient facing materials of the trial.

## Stage-1: non-randomised multicentre learning and evaluation stage (IDEAL stage IIB trial)

Stage-1 is a non-randomised, multicentre learning and harmonisation stage in which quality control measures and mentoring will be employed, to improve and evaluate standards of practice based on the principles of an IDEAL Framework stage-IIB study.[30 31] It will evaluate standard care surgery with the addition of DTI imaging and the ultrasound imaging during the operation. This will ensure that the surgeons using the technologies to be employed in the RCT demonstrate acceptable expertise

in delivering the new approach prior to proceeding with the randomisation stage. This stage ensures standardisation of the use of the technologies across all trial centres by expert mentoring, and will evaluate quality of delivery, including monitoring of the learning curve for the group as a whole.

Stage-1 is divided into three components:
1. Pre-trial webinar
2. IDEAL stage-IIB (quality assurance, mentoring and trial centres evaluation).
3. End of stage-1, pre-stage-2 RCT, each participating centre will have a data workflow review with the lead investigators to review the cases completed in stage-1.

The IDEAL Framework stage-IIB trial will comprise the following:

► Mentoring for local site surgeons.
► Quality assurance of operative procedure.

Mentoring by the chief investigator and lead investigators will be provided through visits to participating centres and frequent meetings, together with a helpline for individual advice sessions from the chief investigator and lead investigators and co-applicants, as appropriate. Neurosurgeons will contribute data to ensure standardisation of the protocol and acceptable expertise in delivering the new approach (online supplemental files 1 and 2). This will be evaluated using the following metrics: operation length; successful use of DTI neuronavigation and iUS to achieve maximal safe tumour resection without major neurological deficit; and extent of tumour resection assessed on postoperative MRI scan. The number of cases required for this may vary but is expected to be small (up to five cases) as most surgeons are already familiar with the component techniques and are not anticipated to require substantial assessment. Ensuring all participating surgeons are ready to take part will minimise performance bias in stage-2 and ensure standardisation of intraoperative technique.

### Stage-2: prospective, stage-III, multicentre RCT with internal pilot

This is a parallel group, two arm, multicentre, RCT. Patients who agree to take part this trial will be allocated by chance. The trial will enrol 357 newly diagnosed patients with GB and will randomly allocate them to receive either surgical resection with standard methods without ultrasound and DTI, or this surgery with the addition of ultrasound and DTI, as well as standard tools. Patients will not know into which group they have been placed, nor will the research team assessing them before and after surgery. Patients will be recruited from at least 15 National Health Service (NHS) hospitals that routinely undertake GB surgery and have access to these tools. The trial will be embedded within existing care pathways. After agreeing to take part, participants will be asked to complete questionnaires (online supplemental files 3-7) about their HRQoL, reflecting symptoms as well as physical, emotional and psychological functioning. They will also have a brief physical and cognitive/functional assessment before their surgery. Afterwards, the questionnaires and assessments will be repeated, before leaving hospital, and at 3 monthly intervals until 24 months after randomisation. These will be combined with planned hospital visits. OS will also be recorded. See figure 1 for a flowchart of the trial.

### Population

Three hundred and fifty-seven participants with GB suitable for maximal, safe resective surgery (attempted GTR of all enhancing tumour), as agreed at the local neuro-oncology multidisciplinary team (MDT) meeting.

### Intervention

Standard care surgery (neuronavigation based on preoperative imaging and intraoperative use of 5-ALA) with the addition of DTI neuronavigation and iUS.

### Control

Standard care surgery (neuronavigation based on preoperative imaging and intraoperative use of 5-ALA)

### Outcome

Deterioration free survival, defined as the time to a 10-point deterioration in HRQoL scores from baseline, without subsequent 10-point improvement in scores compared with baseline; or progressive disease; or death in the absence of previous definitive deterioration before the next assessment. HRQoL is measured with the European Organisation For Research And Treatment Of Cancer (EORTC) Core Quality of Life questionnaire (QLQ-C30) and Brain tumour module (QLQ-BN20) questionnaires.

### Setting

At least 15 UK NHS Trusts undertaking GB surgery

### Eligibility

Patients aged 18–70 years with a primary GB tumour which is deemed maximally resectable (attempted GTR of all enhancing tumour) by the local neuro-oncology MDT meeting, will be potentially suitable for inclusion in the trial.

### Inclusion criteria

► Age 18–70 years
► Neuro-oncology MDT decision that the imaging shows a primary GB tumour which is maximally resectable (attempted GTR of all contrast-enhancing tumour).
► Patient is suitable for concomitant adjuvant radiotherapy and temozolomide (TMZ) chemotherapy followed by adjuvant TMZ at the time of MDT decision.
► Able to receive 5-ALA
► Willing and able to give informed consent.
► Able to complete trial questionnaires, this may be with support where English is not their first language (where compatible with the validation of questionnaires) (stage-2 only).
► Able to provide a proxy who is willing to complete questionnaires as requested (stage-2 only).

### Exclusion criteria

The participant may not enter the trial if any of the following apply:
► Midline/basal ganglia/cerebellum/brainstem GB
► Multifocal GB
► Recurrent GB
► Suspected secondary GB
► Contraindication to MRI

## Proxy inclusion (stage-2 only)

Although it is widely recognised in HRQoL research that an individual may rate aspects of their functioning and well-being differently from how another person might, even if that person is close to the individual (eg, carer, partner), we will ask proxies to also rate HRQoL aspects of patients during the RCT.

The proxy/participant assessment is particularly important in cases where the patients are not able to complete the questionnaires, for example, if they have disease progression, or if their condition is too poor. These proxy measures can be used as substitute data in case the patient's rating of their HRQoL is lacking. When a participant dies, loses capacity or withdraws from the trial—this will also automatically cease the proxy's involvement in the trial.

### Inclusion criteria for proxy

▸ Age 18–75 years.
▸ Nominated by an individual who has consented to participate in stage-2.
▸ Willing and able to give informed consent.
▸ Able to understand written English to enable completion of trial questionnaires.

### Recruitment

Recruitment into the trial will be undertaken in two phases in conjunction with the separate stages of the trial. There will be a separate patient information sheet and consent form for patients entering stage-1 (IDEAL IIb) and stage-2 (RCT) (online supplemental files 8-13). The stages are sequential at participating sites and the stages cannot be recruited in parallel.

All potentially eligible participants will have the trial mentioned at the same time the options regarding their surgery are discussed. Depending on the site, the resources available and most importantly how the participant is dealing with their diagnosis, the recruitment process and approach may vary across and within sites. Potential participants may straight away be provided with the trial participant information sheet and asked to consider the trial, and that a member of the local research team will contact them. It may be the case that individuals are asked if it would be acceptable for their name to be passed to the research team who will make contact at a later time point, or potential participants may be given the participant information sheet and asked to call the number on it if they wish to find out more about the trial.

### Randomisation

Randomisation of patients will only occur in stage-2 of the trial. Every centre and each participating surgeon will offer surgery under both arms of the trial. Randomisation will be via the web-based service provided by Oxford Clinical Trials Research Unit (OCTRU), using the method of minimisation. The minimisation factors will be trial site, age (≤55 years or >55 years), expected surgery status (under general anaesthesia or awake), and eloquence of tumour location (non-eloquent or eloquent).

Participants will be randomised on a 1:1 basis, after having given written consent; however, they will remain blinded as to which arm of the trial they have been allocated. The local clinical team at site will receive an email from the randomisation system detailing the arm of the trial to which a participant has been randomised. Randomisation must occur before the preoperative imaging takes place so that the assigned trial pre-operative imaging can be undertaken.

### Pre and post randomisation withdrawals

Participants may decline to continue to take part in the trial at any time without prejudice. A decision not to participate or withdraw will not affect the standard of care the patient receives. Once withdrawn, the patient will be advised to discuss their further care plan with their surgeon. On withdrawal of the patient, any data collected up until the time of withdrawal will be retained by the research team and included in the final analysis.

### Blinding

Stage-1 is not blinded; the participants will be receiving all the technologies during their surgery.

In stage-2, the participant will be blinded to the allocation (intervention or control arm), and the treating clinician will be aware of the need to perform the surgery with the intraoperative technologies as allocated. In addition to the participant, the radiologist (reviewing the postoperative MRI) will be blinded to the trial arm. Given this, only on the operation case report form (CRF) will data of the allocation be included.

### Trial treatments

All participants will undergo surgery for removal of their GB. The choice of anaesthesia will be left to the discretion of the treating surgeon/anaesthetist/patient as per their normal practice and preference.

The trial will compare two imaging techniques for imaging the tumour. Participants will be randomised to either:
▸ Standard care surgery (neuronavigation based on preoperative imaging and intraoperative use of 5-ALA) (control arm).
▸ Standard care surgery (neuronavigation based on preoperative imaging and intraoperative use of 5-ALA) **AND** of DTI neuronavigation and iUS (intervention arm).

### Objectives and outcome measures

Objectives and outcome measures are summarised in table 1.

### End of trial

The end of the trial will be defined as the collection/receipt of the last follow-up questionnaire from the last participant and all data cleaning has been completed.

**Table 1** Objectives and outcome measures for the trial

| | Objectives | Outcome measures | Time point(s) |
|---|---|---|---|
| **Stage 1** | | | |
| Primary outcomes | To demonstrate the feasibility of using DTI and iUS* in addition to standard of care (neuronavigation based on preoperative MRI and intraoperative use of 5-ALA) for neurosurgery (at selected UK NHS hospitals). | 1. Operation length.<br>2. Successful use of DTI neuronavigation and iUS* to achieve maximal safe tumour resection without major neurological deficit.<br>3. Extent of tumour resection assessed on postoperative MRI scan.<br>4. Surgical complication and serious adverse events | Hospital discharge and 6 months post-op. |
| **Stage 2** | | | |
| Primary outcomes | To assess whether additional intraoperative imaging (DTI and iUS*) to standard of care (neuronavigation based on preoperative MRI scan and intraoperative 5-ALA) improves deterioration free survival (DFS) (where deterioration relates to global health status only). | Composite of the global health status domain of the QLQ-C30 questionnaire, progression free survival (PFS) and overall survival (OS) with an event defined as either deterioration, progression or death. | To be recorded at baseline; 6 weeks post-op., 3 months post-op. and then 3 monthly up to 24 months. |
| Secondary outcomes | To assess whether additional intraoperative imaging (DTI and iUS*) to standard of care (neuronavigation and intraoperative 5-ALA) improves DFS where deterioration relates to physical functioning, social functioning from the QLQ-C30 and motor dysfunction and communication deficit. | Four composites using the respective domain of QLQ-C30 (physical functioning and social functioning) and BN20 (motor dysfunction and communication deficit) combined with PFS and OS. | To be recorded at baseline; 6 weeks post-op., 3 months post-op. and then 3 monthly up to 24 months. |
| Secondary outcomes | To assess whether additional intraoperative imaging (DTI and iUS*) to standard of care (neuronavigation and intraoperative 5-ALA) improves time to deterioration. | Defined similarly to DFS with the exception that progression is excluded as an event (ie, only deterioration or death are considered). There will be five time to deterioration outcomes, one for each of the domains used in the primary and secondary DFS outcomes, used in turn to define deterioration. | To be recorded at 6 weeks post-op., 3 months post-op. and then 3 monthly up to 24 months. |
| Secondary outcomes | To assess whether additional intraoperative imaging (DTI and IUS*) to standard of care (neuronavigation and intraoperative 5-ALA) improves OS. | OS (time from randomisation to death or trial closure). | To be recorded at 24 months. |
| Secondary outcomes | To assess whether additional intraoperative imaging (DTI and iUS*) to standard of care (neuronavigation and intraoperative 5-ALA) improves PFS. | PFS (time from randomisation to radiological tumour progression on imaging, as agreed in local MDT. | MRI at 6 months post-op., and then 3 monthly up to 24 months or an MRI performed outside protocol if patient is symptomatic. |
| Secondary outcomes | To assess whether additional intraoperative imaging (DTI and iUS*) to standard of care (neuronavigation and intraoperative 5-ALA) improves the extent of tumour resection. | Extent of resection as volume of residual tumour postoperative contrast-enhanced MRI.<br>Extent of resection as percentage of preoperative tumour volume on postoperative contrast-enhanced MRI. | Postoperative review. |
| Secondary outcomes | To assess whether additional intraoperative imaging (DTI and iUS*) to standard of care (neuronavigation and intraoperative 5-ALA) improves the incidence of surgical complications. | Number and type of surgical complications. | To be recorded at 5 days post-op, or discharge date (whichever is soonest); 6 weeks post-op., 3 months post-op. and then 3 monthly up to 24 months. |
| Secondary outcomes | To assess whether additional intraoperative imaging (DTI and iUS*) to standard of care (neuronavigation and intraoperative 5-ALA) improves the number of patients eligible for adjuvant treatment following surgery. | Number of patients eligible for adjuvant treatment. | 3 months post-op. |

**Table 1** Continued

|  | Objectives | Outcome measures | Time point(s) |
|---|---|---|---|
| Secondary outcomes | To assess whether additional intraoperative imaging (DTI and iUS*) to standard of care (neuronavigation and intraoperative 5-ALA) improves functional outcome postoperatively. | WHO performance status mini-MoCA (Montreal Version) Barthel Index MRC grading of power in all four limbs. | To be recorded at baseline, 5 days post-op., or discharge date (whichever is soonest); 6 weeks post-op., 3 months post-op. and then 3 monthly up to 24 months. (MRC grading to be assessed at baseline and 5 days post-op., or discharge date only). |
| Secondary outcomes | Assess the correlation of proxy to participant classification assessment of quality of life. | At a minimum, answers to questions 29 and 30 of the QLQ-C30. Ideally answers will be provided to all of the QLQ-C30 and BN20. | Baseline, 6 weeks post-op., 3 months post-op. and then 3 monthly up to 24 months. Proxy will not complete questionnaires when the participant stops completing them. |
| Tertiary mechanistic study objectives— on a subset of participants— | To assess the sensitivity and specificity of the anatomico-spatial location of DTI fibre tracts compared with intraoperative direct electrical stimulation/behavioural change without stimulation but related to adjacent white fibre tract in patients undergoing awake surgery, or motor evoked potential changes in patients undergoing surgery. | Sensitivity and specificity calculation using pre-surgery and post-surgery MRI images. | Analysis will be undertaken post-surgery. |
| Tertiary mechanistic study objectives— on a subset of participants— | To assess the sensitivity and specificity of iUS* to identify the tumour boundary when compared with 5-ALA, navigated biopsies will be taken from tumour boundary tissue planned for resection. | Intraoperative iUS* images and postoperative MRI scans and intraoperative biopsy samples. | Analysis will be undertaken post-surgery when biopsy results are available. |

*if NiUS available, it is to be used.

5-ALA, 5-aminolevulinic acid; BN20, Brain Neoplasm 20 question module; DTI, diffusion tensor imaging; iUS, Intraoperative ultrasound; iUS, intraoperative ultrasound; MDT, multidisciplinary team; MoCA, Montreal Cognitive Assessment; MRC, Medical Research Council; NHS, National Health Service; post-op, postoperative; QLQ-C30, Quality of Life Questionaire - Core 30 module.

## Statistical methods

Full details of the statistical analysis will be detailed in a separate statistical analysis plan which will be drafted early in the trial and will be finalised after input from the trial steering committee (TSC) and data and safety monitoring committee (DSMC). A summary of the planned statistical analysis is included here.

The analysis of the primary outcome will be a time-to-event analysis using a mixed effect Cox proportional hazard regression model. Minimisation factors (age, anticipated patient operative state and tumour location), radiotherapy and methylguanine-DNA methyltransferase status will be adjusted for as fixed effects. Centre will be included as a random effect.

The assumption of proportional hazard for the Cox model will be examined. If the proportional hazard assumption is not met, parametric survival analysis, such as the accelerated failure time method will be considered. A sensitivity analysis will look at the impact of adjusting for the surgeon instead of the centre. Secondary analysis will explore the influence of progression as an event by assessing DFS minus disease progression as an event. An unadjusted comparison using a log-rank test will also be carried out. Kaplan-Meier curves will also be generated.

Secondary time-to-event outcomes (eg, OS) will be analysed in a similar manner.

HRQoL among survivors will be quantified without a formal statistical comparison between treatment groups.

There are multiple factors that may influence how a patient rates their level of HRQoL, which may be related to factors other than the intervention. However, by using a randomised trial design, it is assumed that patients in both treatment arms are comparable on all aspects, both measured (eg, age, performance status) and unmeasured (eg, mood, coping strategy, personality). This means that the impact of the psychological state on the evaluation of HRQoL is treated as similar for the two trial arms. Thus, the trial will be able to measure whether the experimental intervention has an impact on HRQoL when compared with patients receiving standard treatment.

The trial will attempt to collect data as completely as possible. The main analysis will include participants for whom endpoint data are available, with other participants being censored after their last available relevant outcome measure. Sensitivity analyses will examine the effects of alternative assumptions about the missing data. Further details will be provided in the statistical analysis plan, and the data monitoring plan.

## Sample size determination for IDEAL framework IIB trial (stage-1)

There is no formal sample size for the IDEAL trial. Participants will be recruited at each centre, the number of cases required from each centre will vary depending on caseload numbers and the number of neurosurgeons but is expected to be small for most sites (five), as the participating centres are already familiar with the component techniques.

## Sample size determination for the RCT (stage-2)

The sample size is based on a HRQoL aspect included in the primary outcome DFS, that is, the global health status domain in the EORTC QLQ-C30 questionnaire V.3.0, and achieving a statistical power of 90% for the primary analysis with two-sided significance level of 5%. Assuming a HR of 0.7, median DFS survival time of 5 months in the control arm, 24 months follow-up on all participants and allowing for 5% loss to follow-up occurring by month 3, this yields an overall target of 357 participants (178/179 per arm; 335 events overall) (Stata 'artsurv'; www.stata.com). In a recent trial, the mean survival time of global health status DFS was 6 months in the standard treatment arm (surgical resection with standard radiotherapy and chemotherapy).[32] Additionally, the observed HR was 0.64, 95% CI (0.56 to 0.74) for the DFS measures in this trial suggesting that a HR of 0.7 as assumed above is a plausible magnitude of effect to be observed for this population.[33] It would also be the one which would be considered important to clinicians and patients given the definition of a DFS event (death, progression or a patient anchor determined clinically meaningful deterioration of 10 points). For key secondary outcomes (ie, the other four DFS outcomes, PFS and OS) there is over 80% power for this size of trial, assuming a median OS of 6–9, 7 and 15 months, respectively, in the control arm, a HR of 0.70 for both, and other inputs as per above.

## Decision points

### Stage-1 (IDEAL IIB trial)

The trial team will evaluate patient CRF and imaging data continuously on a case-by-case basis from each site and provide regular feedback and assessment. Any additional training/guidance is provided as needed. After a site has done an adequate number of cases and has objectively met the primary outcomes and workflow requirements, the completed data set will be re-evaluated by the trial team including the chief investigator and lead investigators. A meeting between the trial team and the site is then held to allow feedback from the site and discussion of lessons learnt. This meeting is formally documented and if all the criteria are met, the site can then progress to stage 2 (online supplemental files).

### Stage-2 (RCT)

Built into the trial is an internal pilot of recruitment to the RCT (stage-2). There will be a formal stop-go review after 12 months of recruitment to the RCT to review the number of randomisations over the pilot period—the stop-go criteria are listed in table 2. If the target of at least 80 randomisations has been met, the trial will continue to recruit for a further 15 months. Data from the 80 patients will be included in the final analysis.

## Data management

Data will be collected from participants and proxies via questionnaires and CRFs that will be returned to the central trial office in Oxford, via post using a pre-addressed freepost envelope, NHS email as appropriate or directly into an online secure database (REDCap—Research Electronic Data Capture). In addition, participant images will be stored within the cloud database Quentry (Brainlab). As a third-party processor, Brainlab will not receive any data that could identify participants.

All trial-specific documents, except for the signed consent form and follow-up contact details, will refer to the participant with a unique trial participant number/code and not by name. The data will be stored and used in compliance with the relevant, current data protection laws (Data Protection Act 2018; General Data Protection Regulation 2018). The trial data (including data for serious adverse events (SAEs)) will be entered onto a validated REDCap trial database developed and maintained by OCTRU and which can only be accessed by authorised users via the application. After closure of the trial and data analyses, the data will be made publicly available

**Table 2** Proposed stop-go criteria for the TSC at 12 months

| Target=80 | Actual recruitment after 12 months of recruitment | | |
| --- | --- | --- | --- |
| | >80 participants | 65–80 participants | <65 participants |
| Recruitment rate (per centre per month) | 0.6 | 0.45 | 0.37 |
| Stop-go criteria | Recruitment feasible. | Review recruitment strategies. | Recruitment not feasible. |
| | Proceed with trial. | Report to TSC. | Decision not to proceed. |
| | | Continue but modify and monitor closely. | |

TSC, trial steering committee.

at the time of publication. The trial master file will be archived for 5 years from the end of the trial.

## Patient and public involvement

The trial focuses on keeping good HRQoL for people living with a GB for as long as possible. It has been designed with the help of patient support groups at The Brain Tumour Charity and brainstrust, the Patient Relative Advisory Group at the Oxford University Hospitals NHS Foundation Trust and the Brain Tumour PPI Group at Imperial College Healthcare NHS Trust. Dr Helen Bulbeck (brainstrust's director) has been part of the trial proposal and is one of the trial's investigators.

## Trial oversight

The day-to-day management of the trial will be the responsibility of the clinical trial manager, based at Nuffield Department of Surgical Sciences and supported by the OCTRU and the Surgical Intervention Trials Unit staff all based at the University of Oxford with the chief investigator. This will be overseen by the trial management group, who will meet monthly to assess progress.

A TSC and a DSMC will be set up. The DSMC will adopt a DAMOCLES (DAta MOnitoring Committees: Lessons, Ethics, Statistics) based charter which defines its terms of reference and operation in relation to oversight of the trial. They will not be asked to perform any formal interim analyses of effectiveness. They will, however, see copies of data accrued to date and summaries of that data by treatment group. They will also consider emerging evidence from other related trials or research and review-related SAEs that have been reported. They may advise the chair of the TSC at any time if, in their view, the trial should be stopped for ethical reasons, including concerns about participant safety. DSMC meetings will be held at least annually during the recruitment phase of the study.

## Quality control

The study may be monitored, or audited in accordance with the current approved protocol, relevant regulations and standard operating procedures by the host organisation or sponsor. A monitoring plan will be developed according to OCTRU standard operating procedures which involves a risk assessment. The monitoring activities are based on the outcome of the risk assessment and may involve central and site monitoring.

## ETHICS AND DISSEMINATION

The trial was registered in the Integrated Research Application System (Ref: 264482) and approved by a UK research and ethics committee (Ref: 20/LO/0840). Results will be published in a peer-reviewed journal. Further dissemination to participants, patient groups and the wider medical community will use a range of approaches to maximise impact.

### Author affiliations
[1]Department of Neursurgery, Oxford University Hospitals NHS Foundation Trust, Oxford, UK
[2]Nuffield Department of Surgical Sciences, University of Oxford, Oxford, UK
[3]Neurosurgery, Imperial College Healthcare NHS Trust, London, UK
[4]Oxford Clinical Trials Research Unit & Surgical Intervention Trials Unit, University of Oxford Nuffield Department of Orthopaedics Rheumatology and Musculoskeletal Sciences, Oxford, UK
[5]Nuffield Department of Clinical Neurosciences, Oxford University, Oxford, UK
[6]Department of Neurology, Leiden University Medical Center, Leiden, Zuid-Holland, The Netherlands
[7]Department of Neurology, Haaglanden Medical Center Bronovo, Den Haag, Zuid-Holland, The Netherlands
[8]Department of Computer Sciences, UCL, London, UK
[9]Lysholm Department of Neuroradiology, National Hospital for Neurology and Neurosurgery, London, UK
[10]Institute of Cancer and Genomic Studies, University of Birmingham, Birmingham, UK
[11]Department of Neurosurgery, University Hospitals Birmingham NHS Foundation Trust, Birmingham, UK
[12]Brainstrust, Cowes, UK
[13]Institute of Systems, Molecular and Integrative Biology, University of Liverpool, Liverpool, UK
[14]Department of Neurosurgery, Walton Centre for Neurology and Neurosurgery, Liverpool, UK
[15]Department of Clinical Oncology, Imperial College Healthcare NHS Trust, London, UK
[16]Department of Surgery and Cancer, Imperial College London, London, UK
[17]Department of Imaging, Imperial College Healthcare NHS Trust, London, UK
[18]Neuroradiology, Imperial College Healthcare NHS Trust, London, UK
[19]Neurosurgery Division, Department of Clinical Neurosciences, Cambridge University, Cambridge, UK
[20]Department of Neurosurgery, King's College Hospital, London, UK

**Acknowledgements** We would like to thank our participants and other research team members (Amy Taylor, Nadjat Medehgri, Jack Morris, Lucy Eldridge, Ariel Wang and Tianshu Liu) involved in the day-to-day running of the trial. This trial will be conducted as part of the portfolio of trials in the UK Clinical Research Collaboration registered Clinical Trials Unit—the Oxford Clinical Trials Research Unit and the Surgical Intervention Trial Unit at the University of Oxford. It will follow their Standard Operating Procedures ensuring compliance with the principles of Good Clinical Practice and the Declaration of Helsinki and any applicable regulatory requirements.

**Collaborators** FUTURE-GB collaborators: Giles Critchley (Brighton); George Eralil and Kathrin Whitehouse (Cardiff); Anna Solth (Dundee); Paul M Brennan (Edinburgh); Chittoor Rajaraman and Shailendra Achawal (Hull); Anil Varma (Middlesborough); Robert Corns and Ganamurthy Sivakumar (Leeds); Farouk Olubajo (Liverpool); Edward McKintosh, Grainne McKenna and Dimitrios Paraskevopoulos (Barts Health); Neil Barua (Bristol); James Palmer (Plymouth); Stuart Smith (Nottingham); Paul Grundy (Southampton); Erminia Albanese and Huan Chan (Stoke-on-Trent); Damian Holliman (Newcastle); Jose Lavrador and Maria Velicu (King's College Hospital).

**Contributors** PP, SC and DN were responsible for conceptualisation and design of the trial. RM is a trial clinician and drafted the manuscript. JC provided statistical input into the trial design and conduct. CW, MDJ, KA, SJP and VA are trial clinicians and provided input regarding neurosurgical expertise and trial design. MJBT, LDir and MW provided input on neuro-oncology management. MJBT and LDir provided expert opinion on quality-of-life measures. AL and LDix developed section on intraoperative ultrasound techniques and analysis. PM was responsible for guidance on the IDEAL framework. NV and MG-S developed section on diffusion tensor imaging imaging and analysis. HB organised and provided patient and public involvement input. VSB provided input into the trial design. All authors provided critical appraisal of the protocol and the manuscript.

**Funding** This paper presents independent research funded by the National Institute for Health Research (NIHR) under its Efficacy and Mechanism Evaluation Programme (Grant Reference: NIHR 127930; https://fundingawards.nihr.ac.uk/award/NIHR127930).

**Disclaimer** The views expressed are those of the author(s) and not necessarily those of the NIHR or the Department of Health and Social Care. The University of Oxford is the sponsor of the trial. The funders and the sponsor of the trial had no explicit role in the trial design.

**Competing interests** None declared.

**Patient and public involvement** Patients and/or the public were involved in the design, or conduct, or reporting, or dissemination plans of this research. Refer to the Methods section for further details.

**Patient consent for publication** Not applicable.

**Provenance and peer review** Not commissioned; externally peer reviewed.

**Open access** This is an open access article distributed in accordance with the Creative Commons Attribution 4.0 Unported (CC BY 4.0) license, which permits others to copy, redistribute, remix, transform and build upon this work for any purpose, provided the original work is properly cited, a link to the licence is given, and indication of whether changes were made. See: https://creativecommons.org/licenses/by/4.0/.

**ORCID iDs**
Ruichong Ma http://orcid.org/0000-0002-4939-8553
Colin Watts http://orcid.org/0000-0003-3531-8791
Stephen John Price http://orcid.org/0000-0002-7535-3009

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
