## [Reviewer comments · BMJ Open]

ARTICLE DETAILS

TITLE (PROVISIONAL)	FUTURE-GB – Functional and Ultrasound guided Resection of Glioblastoma. A two-stage randomised control trial.
AUTHORS	Plaha, Puneet; Camp, Sophie; Cook, Jonathan; McCulloch, Peter; Voets, Natalie; Ma, Ruichong; Taphoorn, Martin J.B.; Dirven, Linda; Grech-Sollars, Matthew; Watts, Colin; Bulbeck, Helen; Jenkinson, Michael; Williams, Matthew; Lim, Adrian; Dixon, Luke; Price, Stephen; Ashkan, Keyoumars; Apostolopoulos, Vasileios; Barber, Vicki; Taylor, Amy; GB, FUTURE; Nandi, Dipankar

VERSION 1 – REVIEW

REVIEWER	Delgado-López, Pedro Hospital Universitario de Burgos
REVIEW RETURNED	07-Jul-2022

GENERAL COMMENTS	The proposed trial is appropriate in terms of methodology and objectives. The only concern is about the variability of iUS guidance usage in each center and its integration with neuronavigation. This variability may add some uncertainty and bias to the treatment arm, which is supposed to be mitigated with the pre-RCT study. This trial is timely and the results will eventually support (or not) the inclusion of routine iUS usage and DTI guidance in GBM surgery.
---

REVIEWER	Peereboom , David Cleveland Clinic, Brain Tumor Center
REVIEW RETURNED	25-Jul-2022

GENERAL COMMENTS	You might consider as a secondary endpoint, the amount of residual tumor after resection as some feel this to be better than extent of resection as an indicator of risk of early progression.
--

VERSION 1 – AUTHOR RESPONSE

Reviewer 1:

Many thanks for your kind comments. We agree there is a certain degree of variability in iUS utility in different centres. As you state, the aim of Stage 1 is to try to standardise this usage. We also feel, that a certain degree of variability in the RCT will also reflect real world usage of iUS.

Reviewer 2:

Many thanks for your insightful comment. We agree with your comment and the amount of residual is already being collected. We will also add it as a secondary outcome.

Many thanks again and we look forward to hearing from you soon.

Yours Faithfully,

Professor Puneet Plaha